# Muscle Infiltration in Chronic Lymphocytic Leukemia: A Diagnostic Challenge

**DOI:** 10.3390/diagnostics15091068

**Published:** 2025-04-23

**Authors:** Jiro Ichikawa, Keita Kirito, Tomonori Kawasaki, Kojiro Onohara, Masanori Wako, Hirotaka Haro

**Affiliations:** 1Department of Orthopaedic Surgery, Interdisciplinary Graduate School of Medicine, University of Yamanashi, Yamanashi 409-3898, Japan; wako@yamanashi.ac.jp (M.W.); haro@yamanashi.ac.jp (H.H.); 2Haematology and Oncology, Interdisciplinary Graduate School of Medicine, University of Yamanashi, Yamanashi 409-3898, Japan; kirito@yamanashi.ac.jp; 3Department of Pathology, Saitama Medical University International Medical Center, Saitama 350-1298, Japan; tomo.kawasaki.14@gmail.com; 4Radiology, Interdisciplinary Graduate School of Medicine, University of Yamanashi, Yamanashi 409-3898, Japan; konohara@yamanashi.ac.jp

**Keywords:** chronic lymphocytic leukemia, muscle infiltration, ibrutinib, imaging, inflammation

## Abstract

Chronic lymphocytic leukemia (CLL) is the most common leukemia in adults but is rare in Asia. Extramedullary and extranodal manifestations in CLL are generally uncommon, and muscle involvement is extremely rare. A 70-year-old male with CLL presented with bilateral plantar pain, predominantly on the left side. Anemia and reduced platelet count prompted ibrutinib treatment. MRI revealed high-signal areas in the muscles, suggesting inflammation. Anemia and thrombocytopenia improved, but the pain persisted for 8 months. Histopathological findings confirmed CLL infiltration of the muscles. Radiotherapy alleviated the pain, and the patient remains under observation. Careful caution was needed because (1) MRI findings suggested an inflammatory lesion, broadening differential diagnosis, and (2) CLL may coexist with inflammatory diseases. Histopathological examination is essential for correct diagnosis and treatment.

**Figure 1 diagnostics-15-01068-f001:**
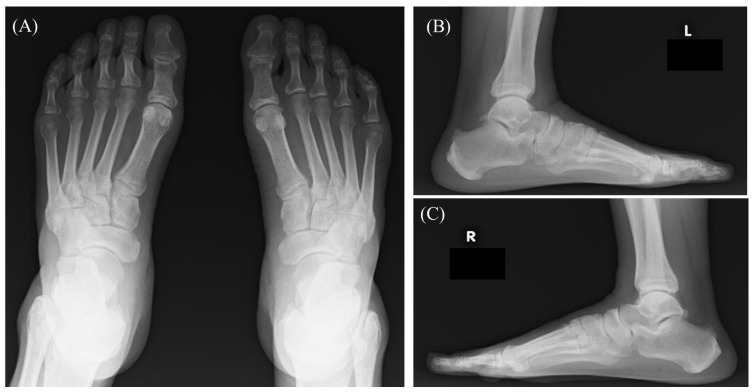
A 70-year-old male with chronic lymphocytic leukemia (CLL) presented with bilateral plantar pain, predominantly on the left side. Nine years ago, due to the presence of leukocytosis (white blood cell count 17,400/uL), the patient was referred to our hospital. Flow cytometry of peripheral blood confirmed the presence of a CD5-positive B-cell population expressing CD19, CD20, and CD23, with no *IgH*/*CCND1* rearrangement detected by fluorescent in situ hybridization. Taken together, a diagnosis of CLL was made. At that time, only bilateral cervical lymph nodes were palpable. Laboratory tests showed hemoglobin at 15.9 g/dL, and the platelet count was 151 × 10^9^/L, suggesting an intermediate risk in the Rai staging system and Stage A in the Binet staging system. Therefore, the criteria for starting treatment were not met. Around the time of plantar pain onset, computed tomography revealed generalized lymphadenopathy and hepatosplenomegaly. In addition, the patient experienced worsening anemia (hemoglobin 10.1 g/dL) and reduced platelet count (88 × 10^9^/L), prompting ibrutinib treatment. X-ray revealed no significant joint deformity, bone destruction, or calcification, with no differences between the right and left sides (**A**–**C**).

**Figure 2 diagnostics-15-01068-f002:**
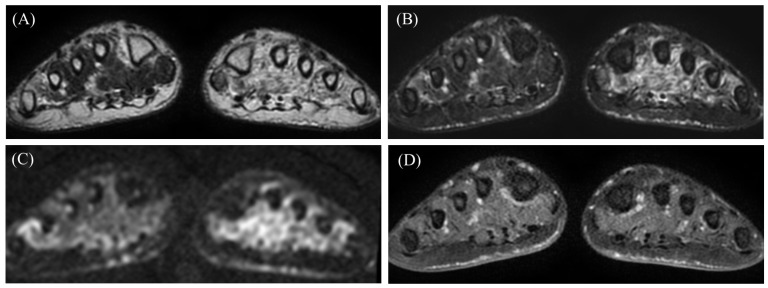
Magnetic Resonance Imaging (MRI) of the feet revealed high-signal areas in the interosseous and plantar muscles, predominantly on the left side, on T2-weighted MRI (**A**) and fat-suppressed T2-weighted MRI (**B**). The high-signal areas on T2-weighted MRI and fat-suppressed T2-weighted MRI matched those on diffusion-weighted MRI (**C**). There is minimal contrast enhancement (**D**). The muscle structure is preserved with no clear mass formation, and there is slight fluid accumulation in the tendon sheath. The findings suggest an inflammatory lesion and denervation, rather than a tumor.

**Figure 3 diagnostics-15-01068-f003:**
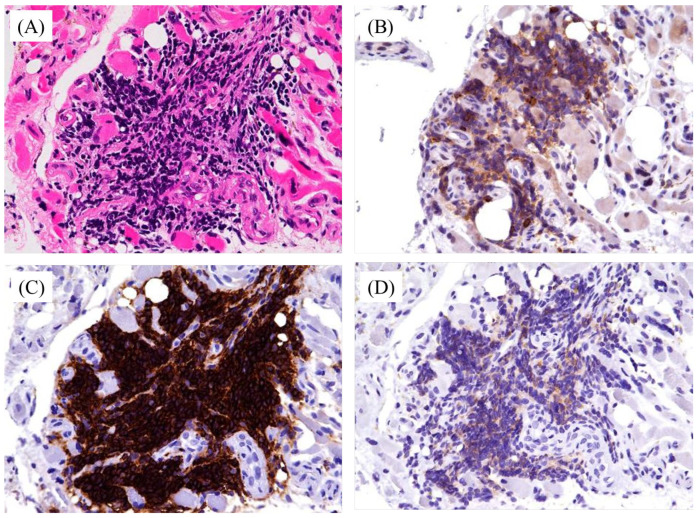
Anemia and thrombocytopenia were improved after the initiation of ibrutinib; however, plantar pain persisted for approximately eight months despite the use of nonsteroidal anti-inflammatory drugs. An incisional biopsy revealed striated muscle fibers with atrophy and degeneration, with small to medium-sized mature lymphocytes, which were hyperchromatic and infiltrating in clusters (**A**). Immunohistochemistry showed positivity for CD5 (**B**), CD20 (**C**), CD23 (**D**) and negativity for CD43, cyclin D1, SOX11. The final diagnosis of CLL muscle infiltration was made. Subsequently, radiotherapy was performed locally and bilaterally, resulting in pain relief, and the patient remains under observation. CLL is the most common leukemia in adults, though it remains relatively rare in Asia, including Japan [1]. A typical feature of CLL is the clonal proliferation and accumulation of mature B-cells within the blood, bone marrow, lymph nodes, and spleen [1]. Neoplastic B-cell diseases, such as monoclonal B-cell lymphocytosis and chronic lymphocytic leukemia/small lymphocytic lymphoma, fall under the same category [1]. In general, the diagnosis of CLL is established through blood counts, differential counts, blood smear, and immunophenotyping. In immunophenotyping, CLL cells co-express the surface antigen CD5 together with the B-cell antigens CD19, CD20, and CD23 [1]. However, caution is needed when interpreting CD5 expression, as it is also observed in other lymphoid malignancies, such as mantle cell lymphoma. For the treatment, several kinds of agents were used: (1) cytostatic agents, (2) monoclonal antibody including CD20, CD52, (3) signaling pathway targeted agents for PI3K, BTK, and BCL-2, and (4) checkpoint inhibitors for PD-1. In particular, BTK plays a pivotal role in B-cell survival pathways downstream of NF-kB and MAP kinases, leading to therapeutic effectiveness. Combination therapy has also been recommended [1]. Extranodal and extramedullary involvement in CLL can affect organs such as the liver, lungs, kidneys, gastrointestinal tract, bone, prostate, and heart. The frequency of skin, cardiac pulmonary involvement was reported as 3–5%, 1.3%, and 5%, respectively. Muscle lesions in CLL are extremely rare, and to our knowledge, this is one of the few reported cases of such involvement. The relationship between extranodal involvement and prognosis remains debatable, with general CLL agents used for treatment [2]. In our case, ibrutinib improved anemia and thrombocytopenia, but the plantar pain persisted, necessitating additional therapy. Radiation therapy was chosen due to its established effectiveness in treating cutaneous CLL [3], with 15 of 19 cases showing a response to radiation and 8 cases exhibiting extended recurrence-free survival during follow-up [4]. Combination chemotherapy with ibrutinib and other agents may be considered as another treatment option. MRI suggested an inflammatory lesion in the muscles, with differential diagnoses ranging from myositis and myopathy to muscular dystrophies, denervation, deep venous thrombosis, diabetic myonecrosis, muscle injury, heterotopic ossification, and even neoplasms [5]. Muscle involvement in NK/T-cell lymphoma has been reported [5,6]. Surprisingly, Stübgen et al. reported that CLL and inflammatory myopathies (IM) may share a bidirectional relationship, although the underlying mechanism remains unclear [7]. IMs represent a diverse collection of muscle diseases driven by immune-mediated inflammation. Key classifications within this group include dermatomyositis (DM), polymyositis (PM), sporadic inclusion body myositis (sIBM), and necrotizing autoimmune myositis [7]. In fact, CLL preceded the development of IM by up to 10 years, with CLL often discovered during the initial evaluation of IM. Proposed mechanisms for the roles of CLL in IM development include (1) myositis-specific antibodies targeting auto antigens (2) CD5+ B-CLL lymphocytes producing natural autoantibodies with multi-specific binding patterns, (3) CLL-related changes in T-cell populations leading to clonal expansion of cytotoxic CD8+ T-cells, which attack muscle fibers expressing myoantigens in the MHC-1 context, and (4) intramuscular monoclonal B-cells interacting with other immune cells to participate in localized inflammation [7]. Given the potential coexistence of CLL and IM, histopathological findings through biopsy were essential in our case to differentiate CLL muscle infiltration from other inflammatory diseases, as treatment strategies differ significantly. In summary, extramedullary and extranodal manifestations in CLL, although rare, seem to affect various organs, including muscles. CLL may also coexist with inflammatory disease in muscles, suggesting the importance of histopathological findings for correct diagnosis and treatment.

## Data Availability

The data presented in this study are available from the corresponding author upon reasonable request.

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
