# Peer review of "Muscle Infiltration in Chronic Lymphocytic Leukemia: A Diagnostic Challenge"

_diagnostics, 2025, doi:10.3390/diagnostics15091068_

Round 1
Reviewer 1 Report
Comments and Suggestions for Authors
- “Figure1”, “Figure2” and “Figure3” should be written as “Figure 1”, “Figure 2” and “Figure 3”, respectively, or as ““Fig. 1”, “Fig. 2” and “Fig. 3”.
- At the time of bilateral plantar, was there any nodal or visceral organ involvement by imaging study? What was the staging result at initial diagnosis of CLL?
- The 4 panels of Figure 3 should be enlarged, as least the same size as Figure 2.
- The expression of CD23 (Panel 3D) is not convincing at all. How about CD43? Any flow cytometric immunophenotyping? Clonality study?
- Page 3. Line 63. “rich in chromatin” is certainly an incorrect way of expression.
- Page 3. Line 67. “CLL is the most ….”. This should serve as the beginning of a new paragraph.
- There is no need to re-iterate the staging system and treatment options in the text of the latter part of Page 3. Please focus on the particular case.
- Page 3. Line 110 to Page 4. Line 111. “ Surprisingly, Li et al reported that … [7]”. However, reference no. 7 is listed as “ Stübgen, J.P. Inflammatory myopathies and lymphoma. J Neurol Sci 2016, 369:377-389.” This is certainly not written by Li et al.
- Please define “inflammatory myopathy” clearly. Is it used for inflammation of the muscles or tumor infiltration into the muscle fibers such as CLL in this case? The former is reactive/non-neoplastic inflammatory reaction, while the latter indicates tumor infiltration.
- Why the authors consider co-existence of CLL and inflammatory myopathy in this case, rather than pure neoplastic infiltration by CLL tumor cells into muscle fibers?
Professional English editing is mantaory.
Author Response
Dear Reviewers,
We would like to express our sincere gratitude for your valuable feedback and insightful comments on our manuscript. We greatly appreciate the time and effort you have dedicated to reviewing our work , and your suggestions have significantly improved the quality of our study.
Below, we provide detailed responses to each of your comments and outline the revisions made to the manuscript. These revisions are highlighted in red in the manuscript for your reference.
Reviewer1
1.“Figure1”, “Figure2” and “Figure3” should be written as “Figure 1”, “Figure 2” and “Figure 3”, respectively, or as ““Fig. 1”, “Fig. 2” and “Fig. 3”.
Response: Thank you for your feedback regarding the formatting of figure citations. We have made the change accordingly in lines 29, 44, and 71.“Figure1”, “Figure2” and “Figure3” are now revised as “Figure 1”, “Figure 2” and “Figure 3”, respectively, with spaces between “Figure” and the number.
2.At the time of bilateral plantar, was there any nodal or visceral organ involvement by imaging study? What was the staging result at initial diagnosis of CLL?
Response: Thank you for your thoughtful questions. We have added detailed information regarding the initial diagnosis and the appearance of bilateral planter pain in lines 35-39 to clarify these points.
“Taken together, a diagnosis of CLL was made. At that time, only bilateral cervical lymph nodes were palpable. Laboratory tests showed hemoglobin at 15.9 g/dL and a platelet count was 151×10⁹/L, suggesting an intermediate-risk in the Rai staging system and Stage A in the Binet staging system. Therefore, the criteria for starting treatment were not met. Around the time of planter pain onset, computed tomography revealed generalized lymphadenopathy and hepatosplenomegaly.”
No visceral organ involvement was noted at the time of initial diagnosis of CLL. At that time, only bilateral cervical lymph nodes were palpable, and imaging studies showed no evidence of visceral organ involvement.
3.The 4 panels of Figure 3 should be enlarged, as least the same size as Figure 2.
Response: Thank you for your suggestion. We have enlarged the four panels of Figure 3 to match the size of Figure 2, as requested.
4.The expression of CD23 (Panel 3D) is not convincing at all. How about CD43? Any flow cytometric immunophenotyping? Clonality study?
Response: Thank you for your valuable suggestion. We have tested for CD43 and confirmed its negativity by immunohistochemistry (line 76). Additionally, we have included flow cytometric immunophenotyping results in lines 32-34 to further support the analysis.
5.Page 3. Line 63. “rich in chromatin” is certainly an incorrect way of expression.
Response: Thank you for your helpful comment. We have revised the phrase “rich in chromatin” to “hyperchromatic” for a more accurate and precise description of the lymphocytes (line 75).
6.Page 3. Line 67. “CLL is the most ….”. This should serve as the beginning of a new paragraph.
Response: Thank you for your suggestion. We have made the necessary change and moved the sentence “CLL is the most…” to the beginning of a new paragraph, as requested. This revision has been made in lines 79-80.
7.There is no need to re-iterate the staging system and treatment options in the text of the latter part of Page 3. Please focus on the particular case.
Response: Thank you for your suggestion. We have removed the sentences in lines 102-106 of the original manuscript, as they reiterated general staging and treatment options. We have focused more on the particular details of the case, as recommended.
8.Page 3. Line 110 to Page 4. Line 111. “ Surprisingly, Li et al reported that … [7]”. However, reference no. 7 is listed as “ Stübgen, J.P. Inflammatory myopathies and lymphoma. J Neurol Sci 2016, 369:377-389.” This is certainly not written by Li et al.
Response: Thank you for pointing out this mistake. We have corrected the reference in line 108 and updated the author to Stübgen as per the correct citation.
9.Please define “inflammatory myopathy” clearly. Is it used for inflammation of the muscles or tumor infiltration into the muscle fibers such as CLL in this case? The former is reactive/non-neoplastic inflammatory reaction, while the latter indicates tumor infiltration.
Response: Thank you for your suggestion. We have added a definition of “inflammatory myopathy” in lines 110-112 as a diverse group of immune-mediated muscle diseases, which includes conditions like dermatomyositis, polymyositis, sporadic inclusion body myositis, and necrotizing autoimmune myositis.
10.Why the authors consider co-existence of CLL and inflammatory myopathy in this case, rather than pure neoplastic infiltration by CLL tumor cells into muscle fibers?
Response: Thank you for your insightful comment. We considered the co-existence of CLL and inflammatory myopathy in this case due to the possibility of IM associated with CLL, as suggested by the literature. Additionally, MRI findings were consistent with bilateral inflammation, and the histopathological features from the biopsy supported the diagnosis of CLL muscle infiltration along with inflammatory myopathy, rather than pure neoplastic infiltration by CLL tumor cells. These combined findings led us to conclude that both conditions were present in this case.
We appreciate your thoughtful comments and the opportunity to improve the manuscript. We look forward to your feedback on the revised version and would be happy to make any additional changes if needed to further enhance the quality of the work.
Reviewer 2 Report
Comments and Suggestions for Authors
This is an unusual case report which is meritorious if the diagnosis of CLL is confirmed.
Need to do Cyclin D1 or Sox11 to rule out mantle cell lymphoma. The combination of CD5 and CD23 is not fully diagnostic.
Also:
1-Include a figure 2C in focus.
2-Clarify what Patents mean in line 128.
Comments on the Quality of English Language
Acceptable.
Author Response
Dear Reviewers,
We would like to express our sincere gratitude for your valuable feedback and insightful comments on our manuscript. We greatly appreciate the time and effort you have dedicated to reviewing our work , and your suggestions have significantly improved the quality of our study.
Below, we provide detailed responses to each of your comments and outline the revisions made to the manuscript. These revisions are highlighted in red in the manuscript for your reference.
Reviewer2
This is an unusual case report which is meritorious if the diagnosis of CLL is confirmed.
Need to do Cyclin D1 or Sox11 to rule out mantle cell lymphoma. The combination of CD5 and CD23 is not fully diagnostic.
Response: Thank you for your suggestion. We have performed immunohistochemistry for cyclin D1 and SOX11 to rule out mantle cell lymphoma. Both tests returned negative results, confirming that mantle cell lymphoma was not present (line 76).
Also:
1-Include a figure 2C in focus.
Response: This is the original diffusion-weighted MRI image.
2-Clarify what Patents mean in line 128.
Response: Thank you for pointing out this error. The use of the word “Patents” was a mistake, and we have removed it from the manuscript.
We appreciate your thoughtful comments and the opportunity to improve the manuscript. We look forward to your feedback on the revised version and would be happy to make any additional changes if needed to further enhance the quality of the work.
Round 2
Reviewer 1 Report
Comments and Suggestions for Authors
The revision looks nice. The only change that has to be made is in Line 33. IgH/CCND1 refers to genes, and should be written as IGH/CCND1 (italicized capital).
Author Response
Dear Reviewers,
Thank you for your thoughtful feedback and helpful suggestions. We greatly appreciate your time and effort in reviewing our manuscript. Below, we outline our responses to each comment, along with the revisions made, which are highlighted in red in the manuscript.
Reviewer1
1.The revision looks nice. The only change that has to be made is in Line 33. IgH/CCND1 refers to genes, and should be written as IGH/CCND1 (italicized capital).
Response: Thank you for your thoughtful comments. We have made the suggested correction and have italicized "IGH/CCND1" as recommended in Line 33.
Thank you for your valuable feedback. We hope the revisions address your concerns, and we welcome any further suggestions to improve the manuscript.
Reviewer 2 Report
Comments and Suggestions for Authors
Thanks for improving the manuscript. Please revise and spell check the english. For example,
Line 38 planter pain vs plantar.
Also please tone down the claim about muscle lesion in CLL. Karatzanis A, Velegrakis S, Liva G, Kyrmizakis D, Prokopakis E. Management of a Buccal Space Mass: A Clinical Case Report. Case Rep Otolaryngol. 2020 Dec 14;2020:6828453. doi: 10.1155/2020/6828453. PMID: 33457031; PMCID: PMC7787848.
Comments on the Quality of English LanguageSee above.
Author Response
Dear Reviewers,
Thank you for your thoughtful feedback and helpful suggestions. We greatly appreciate your time and effort in reviewing our manuscript. Below, we outline our responses to each comment, along with the revisions made, which are highlighted in red in the manuscript.
Reviewer2
Thanks for improving the manuscript. Please revise and spell check the english. For example, Line 38 planter pain vs plantar.
Response: Thank you for your thoughtful comments. We have made the suggested correction (Lines 38 & 100) and have thoroughly proofread the manuscript with the assistance of Editage to ensure accuracy and clarity.
Also please tone down the claim about muscle lesion in CLL. Karatzanis A, Velegrakis S, Liva G, Kyrmizakis D, Prokopakis E. Management of a Buccal Space Mass: A Clinical Case Report. Case Rep Otolaryngol. 2020 Dec 14;2020:6828453. doi: 10.1155/2020/6828453. PMID: 33457031; PMCID: PMC7787848.
Response: Thank you for your valuable suggestion. We have revised the sentence to tone down the claim, and we now state that “Muscle lesions in CLL are extremely rare, and to our knowledge, this is one of the few reported cases of such involvement” (Line 97-98). We appreciate your input, which helped clarify the statement.
Thank you for your valuable feedback. We hope the revisions address your concerns, and we welcome any further suggestions to improve the manuscript.